# Identification of potent anti-*Cryptosporidium* new drug leads by screening traditional Chinese medicines

**Mohammad Hazzaz Bin Kabir**[1,2,3], **Frances Cagayat Recuenco**[2,4], **Nur Khatijah Mohd Zin**[5], **Nina Watanabe**[1], **Yasuhiro Fukuda**[1], **Hironori Bando**[1,6], **Kenichi Watanabe**[7], **Hiroki Bochimoto**[5], **Xuenan Xuan**[2], **Kentaro Kato**[1,2]*

**1** Laboratory of Sustainable Animal Environment, Graduate School of Agricultural Science, Tohoku University, Osaki, Miyagi, Japan, **2** National Research Center for Protozoan Diseases, Obihiro University of Agriculture and Veterinary Medicine, Obihiro, Hokkaido, Japan, **3** Department of Microbiology and Parasitology, Sher-e-Bangla Agricultural University, Sher-e-Bangla Nagar, Dhaka, Bangladesh, **4** Department of Biology, De La Salle University, Taft Ave., Manila, Philippines, **5** Department of Cell Physiology, The Jikei University School of Medicine, Tokyo, Japan, **6** Asahikawa Medical University, Asahikawa, Hokkaido, Japan, **7** Department of Veterinary Medicine, Obihiro University of Agriculture and Veterinary Medicine, Obihiro, Hokkaido, Japan

* kentaro.kato.c7@tohoku.ac.jp

**Data Availability Statement:** This information will only be available after acceptance.

**Funding:** This study was funded by the Japan Society for the Promotion of Science (JSPS) for

## Abstract

*Cryptosporidium* spp. are gastrointestinal opportunistic protozoan parasites that infect humans, domestic animals, and wild animals all over the world. Cryptosporidiosis is the second leading infectious diarrheal disease in infants less than 5 years old. Cryptosporidiosis is a common zoonotic disease associated with diarrhea in infants and immunocompromised individuals. Consequently, cryptosporidiosis is considered a serious economic, veterinary, and medical concern. The treatment options for cryptosporidiosis are limited. To address this problem, we screened a natural product library containing 87 compounds of Traditional Chinese Medicines for anti-*Cryptosporidium* compounds that could serve as novel drug leads and therapeutic targets against *C. parvum*. To examine the anti-*Cryptosporidium* activity and half-maximal inhibitory doses (EC$_{50}$) of these compounds, we performed *in vitro* assays (*Cryptosporidium* growth inhibition assay and host cell viability assay) and *in vivo* experiments in mice. In these assays, the *C. parvum* HNJ-1 strain was used. Four of the 87 compounds (alisol-A, alisol-B, atropine sulfate, and bufotalin) showed strong anti-*Cryptosporidium* activity *in vitro* (EC$_{50}$ values = 122.9±6.7, 79.58±13.8, 253.5±30.3, and 63.43 ±18.7 nM, respectively), and minimum host cell cytotoxicity (cell survival > 95%). Furthermore, atropine sulfate (200 mg/kg) and bufotalin (0.1 mg/kg) also showed *in vivo* inhibitory effects. Our findings demonstrate that atropine sulfate and bufotalin are effective against *C. parvum* infection both *in vitro* and *in vivo*. These compounds may, therefore, represent promising novel anti-*Cryptosporidium* drug leads for future medications against cryptosporidiosis.

young scientists, Japan (ID No. P21101; JSPS/OF498 to MHBK and KK), Grants-in-Aid for Scientific Research (B:20H03476 to KK) from the Ministry of Education, Culture, Science, Sports, and Technology (MEXT) of Japan and a Livestock Promotional Subsidy from the Japan Racing Association (JRA to KK). The funders had no role in study design, data collection and analysis, decision to publish, or preparation of the manuscript.

**Competing interests:** The authors have no competing interests to declare.

## Author summary

Cryptosporidiosis is a major infectious diarrheal disease and death in children in developing countries. *Cryptosporidium parvum* can cause severe watery diarrhea in infants and immunocompromised individuals. As a result, finding new anti-*Cryptosporidium* medications is a priority. In order to find anti-*Cryptosporidium* compounds, we screened a natural product library containing Traditional Chinese Medicines. We performed a *Cryptosporidium* growth inhibition assay (GIA), and a cytotoxicity assay to reveal the anti-*Cryptosporidium* ability and half-maximal inhibitory concentrations ($EC_{50}$) of the natural products. A host cell viability assay and an *in vivo* experiment were used to determine the compounds toxicity to host cells. Four natural compounds (alisol-A, alisol-B, atropine sulfate, and bufotalin) showed strong anti-*Cryptosporidium* effects and low cytotoxicity (cell viability > 95%) using *C. parvum* HNJ-1 strain. Furthermore, atropine sulfate (200 mg/kg) and bufotalin (0.1 mg/kg) reduced oocyst shedding by 67.8% and 78.1%, respectively. The current study discovered that atropine sulfate and bufotalin had inhibitory effects against *C. parvum* infection *in vitro* and *in vivo*, which had never been previously described. As a result, the chemotherapeutic potential of these compounds are discussed for future anti-*Cryptosporidium* drugs to treat Cryptosporidiosis.

## Introduction

Cryptosporidiosis is present all around the world; however, it is more common in areas with poor sanitation and hygiene. Diarrhea is common in infected immunocompromised or immunosuppressed individuals, and can lead to increased morbidity and mortality, especially among AIDS patients [1]. *Cryptosporidium* parasites are resistant to common disinfectants, including chlorine, making it difficult to eliminate the pathogen [2]. No vaccines are available to prevent the condition, and treatment options are limited, with rehydration therapy being the most common [3].

Only one drug, nitazoxanide, has been approved by the Food and Drug Administration (FDA) to treat cryptosporidiosis [4]. Nitazoxanide remains the most effective current treatment for cryptosporidiosis in immunocompetent individuals, whereas no consistently effective medication exists to treat immunodeficient patients or children under 2 years [5], during HIV coinfection, even long-term NTZ treatment was ineffective [6]. Drug discovery research has recently concentrated on developing novel drugs that are effective against *Cryptosporidium spp*. With a variety of new chemical entities, recent developments in this area provide reason for optimism [7]. Clofazimine also exhibited efficacy against *Cryptosporidium*, making it a potentially new cryptosporidiosis treatment and a novel chemical tool for understanding *Cryptosporidium* biology [8]. In both *vivo* and *in vitro* studies, chitosan, a natural polysaccharide, greatly reduced parasite shedding in infected newborn mice [9]. Bicyclic azetidines currently block *C. parvum* phenylalanyl-tRNA synthetase, allowing for target-based therapeutic development for anticryptosporidial drugs [10]. Several additional chemotherapeutic drugs have been studied in livestock for the treatment of cryptosporidiosis showing efficacy against diarrhea in dairy calves [11–13], however, we need more screening to develop and identify new drug lead compounds to combat this infectious pathogen. Some treatments, such as paromomycin, are effective in preventing *Cryptosporidium* oocyst shedding, clinical disease, and death in calves, lambs, and goat kids, but they are not licensed for use in calves [14]. As a result, the development of new drugs is a priority.

Traditional Chinese Medicines (TCMs), which encompass ancient herbal medicines and wellness practices long embraced in China and its neighboring countries, have been recognized as a model of complementary and alternative medicine that has gained popular international interest and use over time [15]. Compounds in the TCMs are effective against trypanosomiasis [16], malaria [17], toxoplasmosis [18], and cancer [19]. The goal of this study was to find potential new anti-*Cryptosporidium* compounds from a library of 87 different TCMs. Drug screening analysis was performed both *in vitro* and *in vivo*. This screening led to the identification of two compounds that showed growth-inhibitory effects on *Cryptosporidium* both *in vitro* and *in vivo* that could be developed as new drug leads against Cryptosporidiosis.

## Methods

### Ethical approval

All animal experiments described in this study were approved by the Obihiro University of Agriculture and Veterinary Medicine Ethical Committee, and adhered to the relevant laws and regulations on the treatment and use of laboratory animals set forth by the Obihiro University of Agriculture and Veterinary Medicine Regulations for Animal Experiments, Japan, (Animal exp: 20–208; Pathogen exp: 201982), and were approved by the Committee on the Animal Experiments of the Tohoku University (2021171).

### Compound preparation

The Institute of Natural Medicine, University of Toyama, Japan provided the natural drug library used in this study. There are 96 chemicals in this library. In the experiments, 87 readily available chemicals were used. The compounds were dissolved in DMSO and kept at −80˚C at a concentration of 10 mM. Nitazoxanide (Wako, Osaka, Japan) served as standard comparative control for *C. parvum*. All chemical stock solutions were made fresh on the day of the experiment.

### Parasites

Both *in vitro* and *in vivo* research used the same parasite strain. *C. parvum* oocysts, strain HNJ-1 [20], were provided by Dr. Matsubayashi, Osaka Prefecture University, Japan. Oocysts were maintained in experimentally infected SCID mice (CB17/Icr-Prkdcscid/CrlCrlj) and isolated from feces by using discontinuous sucrose as previously described [21] and stored at 4˚C in phosphate-buffered saline (PBS) (pH 7.4).

### *In vitro* cultures of *C. parvum*

*C. parvum* oocysts that were less than 3 months old at the time of collection were used in all tests. The HCT-8 cell line derived from human ileocecal adenocarcinoma (ATCC # CCL-225) was cultured in maintenance medium RPMI 1640 with L-Gln (Nacalai Tesque, Tokyo, Japan) and supplemented with 1 mM sodium pyruvate, 15 mM HEPES, 10% fetal bovine serum (FBS), Penicillin (100 U/ml), Streptomycin (100 μg/ml), Amphotericin B (0.25 μg/ml) solution. Every two days, the cells were passaged *in vitro* at 70–80 percent confluency. The cells were incubated at 37˚C, in 5% $CO_2$.

### *In vitro C. parvum* growth inhibition assay

The primary screening of compounds for *C. parvum* was performed as follows: HCT-8 cells were seeded in 96-well plates at a density of $2 \times 10^4$ cells/well and allowed to grow overnight.

Before infecting cells, *Cryptosporidium* oocysts were excysted to sporozoites. 87 compounds were tested against *C. parvum* at a single concentration of 1 μM. Each compound was tested once. Infected HCT-8 cells and uninfected HCT-8 cells were used as positive and negative controls, respectively, in medium containing 0.1% DMSO. Nitazoxanide was used as a standard comparative drug. *In vitro* inhibition assays were carried out as previously described [22, 23]. Briefly, oocysts were suspended in 1 ml of 0.1% sodium hypochlorite in PBS, incubated for 5 min at 4˚C, centrifuged for 10 min at 3000 rpm, and washed three times in PBS. The oocysts were then suspended in excystation solution consisting of 0.75% sodium taurocholate and 0.25% trypsin dissolved in 0.1 M phosphate buffer (with 93.4 mM $K_2HPO_4$ and 6.5 mM $KH_2PO_4$; pH 8.0) and incubated at 37˚C for 1 h. The oocyst/sporozoite suspensions were then rinsed in 1 x PBS (with 137 mM NaCl, 2.7 mM KCl, 8 mM $Na_2HPO_4$, and 2 mM $KH_2PO_4$; pH 7.5) once more before being filtered through a 5-μm pore-size PVDF filter (Millipore, Burlington, MA, USA) with 5 ml syringe to remove the oocyst wall. HCT-8 cells were seeded in 96-well plates (Thermo Fisher Scientific Inc., MA, USA) and incubated overnight. The cells were then infected with *C. parvum* sporozoites ($4 \times 10^4$ sporozoites/well) containing in RPMI-1640 medium for 3 h. After 3h inoculation, the non-infected parasite or debris were washed with new RPMI-1640 media. At this point, in RPMI-1640 medium containing 1 μM library compounds were added. Infected cells were incubated for an additional 45 h. Following incubation, the infected cells were fixed for 10 minutes in ice-cold 100% methanol at room temperature. Plates were then carefully rinsed 3 times with PBS. The parasite infected cells were then blocked with 1% bovine serum albumin (BSA) in PBS for 30 minutes and then stained with Sporo-Glo (an anti-*Cryptosporidium* polyclonal antibody) (Waterborne Environmental, Inc., VA, USA) protected from light for 1 h at room temperature. Finally, cells were washed twice with 1 x PBS-T (1 x PBS supplemented with 0.01% Tween). A Keyence BZ-II 9000 was used to capture the images (KEYENCE, Osaka, Japan). The plates were imaged with a HS all-in-one fluorescence microscope using a 20 x objective. The image output was imported into Microsoft excel for data organization and analyses. All *C. parvum parasite* in the well were counted. Compounds that showed more than 60% inhibition of *C. parvum* growth relative to the DMSO control were selected as hit compounds.

For a second investigation, four selective hit compounds were used to determine the half-maximum inhibitory concentration ($EC_{50}$). The parasites were treated with various doses of compounds (ranging from 0.008–1 μM). HCT-8 cells ($2 \times 10^4$ cells/well) were grown for 24 h before being infected with $4 \times 10^4$ *C. parvum* sporozoites in 96-well plates. The sporozoites that did not invade were washed with RPMI-1640 3 h after inoculation and treated with the four hit compounds from the original screening at 0.008–1 μM, then incubated for another 45 h. Sporo-Glo was used for *C. parvum* staining as described above. Fluorescence microscopy with a Keyence BZ-II 9000 imager was used to count all *Cryptosporidium* in the wells (KEYENCE, Osaka, Japan). $EC_{50}$ vales were determined by analyzing dose-response curves made by GraphPad Prism (GraphPad Software, CA, USA).

## Host cell viability assay

HCT-8 cells ($4 \times 10^4$ cells/well) 100 μl were seeded in 96-well plates, cultured for 24 h, and then treated for 48 h with different concentrations of hit compounds ($10^1$, 1, $10^{-1}$, $10^{-2}$, $10^{-3}$, $10^{-4}$ μM). After a 48-h incubation period, equilibrate the 96-well plates in room temperature (RT) for 30 minutes. Then, equal volume of Cell-Titer Glo (Promega) reagent was added to each treated well, the plate was shaken on an orbital shaker for 2 minutes then incubated for 10 minutes at RT. The luminescence was quantified by the GloMax Navigator plate reader (Promega, Japan) and the cell viability was evaluated as directed by the kit manufacturer. The

luminescence signal of control wells containing DMSO was set to 100% cell viability. HCT-8 cells were treated with various concentrations of four selective target compounds (ranging from 1.56–200 μM) to determine the half-maximal cellular cytotoxic concentration ($CC_{50}$) values. Graph Pad Prism 7.0.1 was used to determine $CC_{50}$ values.

### *In vivo C. parvum* growth inhibition assay

The growth inhibitory effect of the hit compounds was assessed in a mouse model infected with *C. parvum* oocysts. In this experiment, four-week-old female SCID mice (Charles River Laboratories, Shizuoka, Japan) were used. In the infected groups, mice were orally inoculated with $1.0 \times 10^5$ *C. parvum* oocysts by using sterile gavage needles. Initially, four compounds were tested at different concentrations to figure out the optimum dosages. Six groups of mice were given oral dosages of 50, 25, and 12.5 mg/kg body weight (BW) of two compounds (alisol-A and alisol-B). Another six groups of mice were given oral doses of atropine sulfate (100, 50, and 25 mg/kg BW) and bufotalin (1, 0.5, and 0.1 mg/kg BW, respectively). Based on their survivability, two compounds were chosen: atropine sulfate and bufotalin. Each compound group had three concentrations, and had three mice in each group. One group of mice was orally administered atropine sulfate at doses of 200, 100, and 50 mg/kg BW. Bufotalin was given orally at three different doses to the other mouse group: 0.1, 0.05, and 0.025 mg/kg BW. In another mouse group, nitazoxanide was given orally at 100 mg/kg BW as a standard comparative drug. The positive control group consisted of infected and untreated mice, whereas the negative control group consisted of noninfected and untreated mice. Treatment began on the third day after infection and lasted until the thirteenth day. The body weight and condition were checked every day, and treatment was given based on the body weight. On days 1, 3, 6, 7, 9, 12, 14, fecal samples were taken. The total number of oocysts per gram was calculated after counting the oocysts in the feces using the sugar flotation method. The treated and untreated mice were euthanized by cervical dislocation on day 14, and a thorough necropsy was performed.

### Histopathological analysis and immunohistochemistry

We performed a histopathological analysis to evaluate the growth inhibitory effects and toxicity of the hit compounds in *C. parvum*-infected mice. In brief, ileum was collected, fixed in 10% neutral buffered formalin, and paraffin-embedded. Each section was cut to a thickness of 4 μm for hematoxylin and eosin (HE) staining and immunohistochemistry. Immunohistochemistry (IHC) was performed following standard techniques using the Envision+ system (Agilent Technologies, Inc., Santa Clara, CA) and 3,3-diaminobenzidine (DAB)(Dojindo, Kumamoto, Japan). Anti-Toxoplasma gondii (RH strain) rabbit polyclonal IgG (Abcam, Cambridge, UK) was used as the primary antibody. The antibody cross reacts with *C. Parvum*. To reduce the endogenous peroxidase activity, the sections were incubated with 3% $H_2O_2$ for 5 minutes. Sections were incubated with primary antibodies overnight at 4˚C. After washing with PBS with Tween 20 (PBST), sections were incubated with Envision+ single polymer solution (Agilent Technologies Inc.) for 30 minutes at room temperature. Sections were counterstained with Mayer's hematoxylin. The severity of *C. parvum* infection was evaluated in the most severe area, and scored from—to 3+ (-, no oocysts; 1+, mild focal infection; 2+, moderate multifocal infection; 3+, severe diffuse infection). Sub-effects of drugs (hepatocellular damage) were evaluated for major histological changes in karyomegaly, hepatocellular hypertrophy, and interstitial fibrosis/inflammation. We also checked the organs (liver, heart, lung, kidney, and intestines) alteration at necropsy whether it has abnormal or normal.

## Scanning electron microscopy (SEM) assay

We used SEM to see examine the effects of the chemicals on the tissues of the bowel. Tissue samples from each experimental animal's distal ileum were biopsied and immediately immersed in a fixative combination of 2% glutaraldehyde in 0.1 M phosphate buffer (PB, pH 7.4) for 1 week at 4˚C. The samples were then carefully washed three times with 0.1 M PB and immersed in 1% tannic acid for 2 h at 4˚C. After rinsing with 0.1 M PB, the samples were further treated with 1% osmium tetroxide in 0.1 M PB for conductive staining [24]. The conductive-stained samples were then dehydrated in a succession of increasing ethanol concentrations (70, 80, 90, 95, and 100%), transferred to t-butyl alcohol, and dried in a freeze drier (VFD-21S, Vacuum Device, Ibaraki, Japan). The dried samples were mounted on an aluminum metal plate and osmium coated by an osmium coater (HPC-1SW, Vacuum Device, Ibaraki, Japan). Following the procedure outlined above, the specimens were examined using a field emission SEM (Regulus 8100, Hitachi High Technologies, Tokyo, Japan).

## Statistical analysis

Microsoft Excel was used to calculate the percentage of inhibition. The $EC_{50}$ vales were calculated by using a nonlinear regression sigmoidal dose-response curve fit, available in GraphPad Prism 7.01. (GraphPad Software Inc., USA). The difference in parasitemia between control and drug-treated groups was considered statistically significant for $P < 0.05$ by using a one-way ANOVA with the post-hoc Tukey HSD test.

## Results

### *In vitro* screening of Traditional Chinese Medicines (TCMs)

Initial screening of TCM library products was carried out to identify compounds that inhibit *C. parvum* growth by more than 60% at a concentration of 1 μM. Nine of the 87 compounds tested inhibited *C. parvum* growth (Fig 1 and Table 1); their structures are shown in Fig 2.

### Host cell viability assay

Using HCT-8 cells as a model, we determined the cytotoxicity of these compounds. The eight compounds were tested for their effect on the viability of HCT-8 cells. Alisol-A, alisol-B, atropine sulfate, and bufotalin were selected as target compounds because they inhibited parasite development by a high percentage (72.3%–92.9%) and cell viability was greater than 95% in the presence of these four compounds (Table 1). In addition, the half-maximal cellular cytotoxicity ($CC_{50}$) of these four compounds ranged from 10.9–59.6 μM (Table 2).

### *In vitro C. parvum* growth inhibition

The half-maximal inhibitory concentration ($EC_{50}$) values of alisol-A, alisol-B, atropine sulfate, and bufotalin were found to be 122.9±6.7, 79.58±13.8, 253.5±30.3, and 63.43±18.7 nM respectively (Fig 3). Furthermore, the calculated selectivity index (SI) of alisol-A, alisol-B, atropine sulfate, and bufotalin were 484.4, 460.6, 69.9, and 172.3, respectively, as a measure of therapeutic efficacy (Table 2). These results indicated that these selected four compounds have minimum cytotoxicity in mammalian host cells.

### *In vivo C. parvum* growth inhibition

The chemotherapeutic efficacy of single daily doses of all four compounds at different dose levels was evaluated *in vivo* in the mouse model. Mice treated with alisol-A and alisol-B at 50, 25,

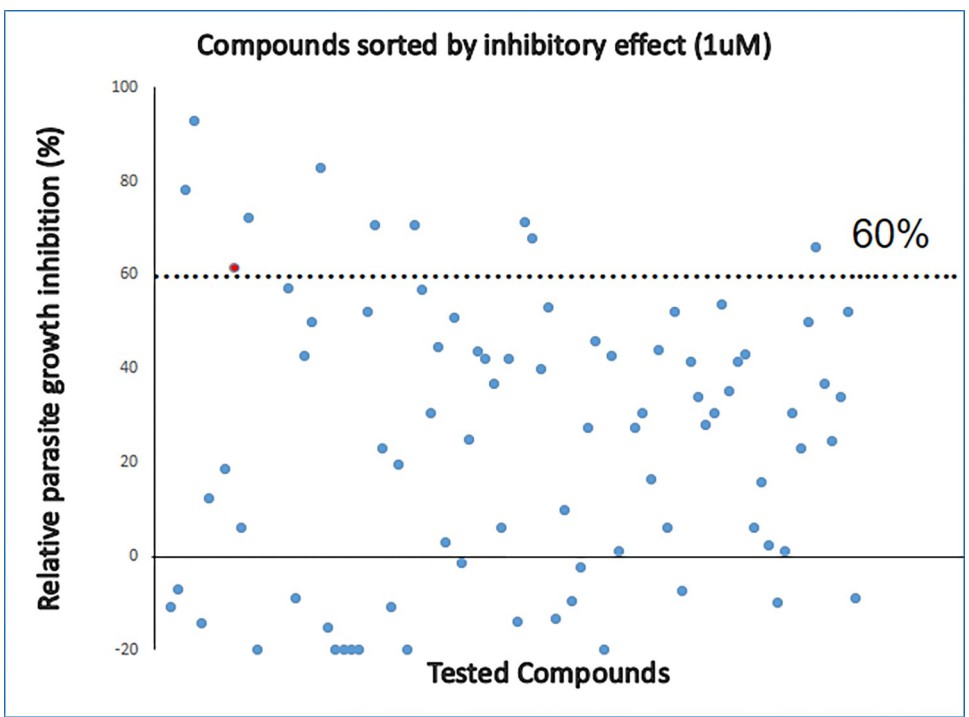

**Fig 1. Scatter plot of tested compounds after primary screening at the single concentration of 1 μM.** The growth of *C. parvum* HNJ-1 strain after exposure to each compound at a single concentration of 1 μM for 45 h, as determined by means of fluorescence microscopy. The percent inhibition is presented as the percentage of inhibited parasites compared with the positive control (untreated wells) after subtraction of the negative control (uninfected HCT-8 cells). Compounds with a growth reduction of more than 60% (dot line) were considered to be compound that inhibit *C. parvum* parasite growth. Blue dots represent the compounds and the red dot represents the comparative drug nitazaxonide (10 μM). This is single experiment with the means of triplicate wells.

and 12.5 mg/kg, respectively, experienced adverse effects and died at concentrations of 50 and 25 mg/kg body weight. Moreover, there was no significant reduction in the numbers of oocysts discharged in the feces of these mice and no significant difference in their body weight (S1 Fig and S1 Table). Both histopathological and scanning electron microscopy findings showed that

**Table 1. The nine lead anti-*C. parvum* compounds identified in the first screen at a concentration of 1 μM and tested against the *C. parvum* HNJ-1 strain.**

| Compound | Parasite inhibition (%) | Host cell viability (%) |
|---|---|---|
| Alisol-A | 78.1 | 99.5 |
| Alisol-B | 92.9 | 97.6 |
| Atropine sulfate | 72.3 | 97.7 |
| Bufotalin | 82.9 | 95.1 |
| Cinobufotalin | 70.7 | 97.8 |
| Dihydrocapsaicin | 70.8 | 98.4 |
| Ginsenoside-Re | 71.2 | 98.7 |
| Ginsenoside-Rg1 | 67.9 | 99.1 |
| Shikonin | 66.2 | NT* |

*NT = Not tested

Parasite growth inhibition values with *C. parvum;* host cell viability values with HCT-8 cells.

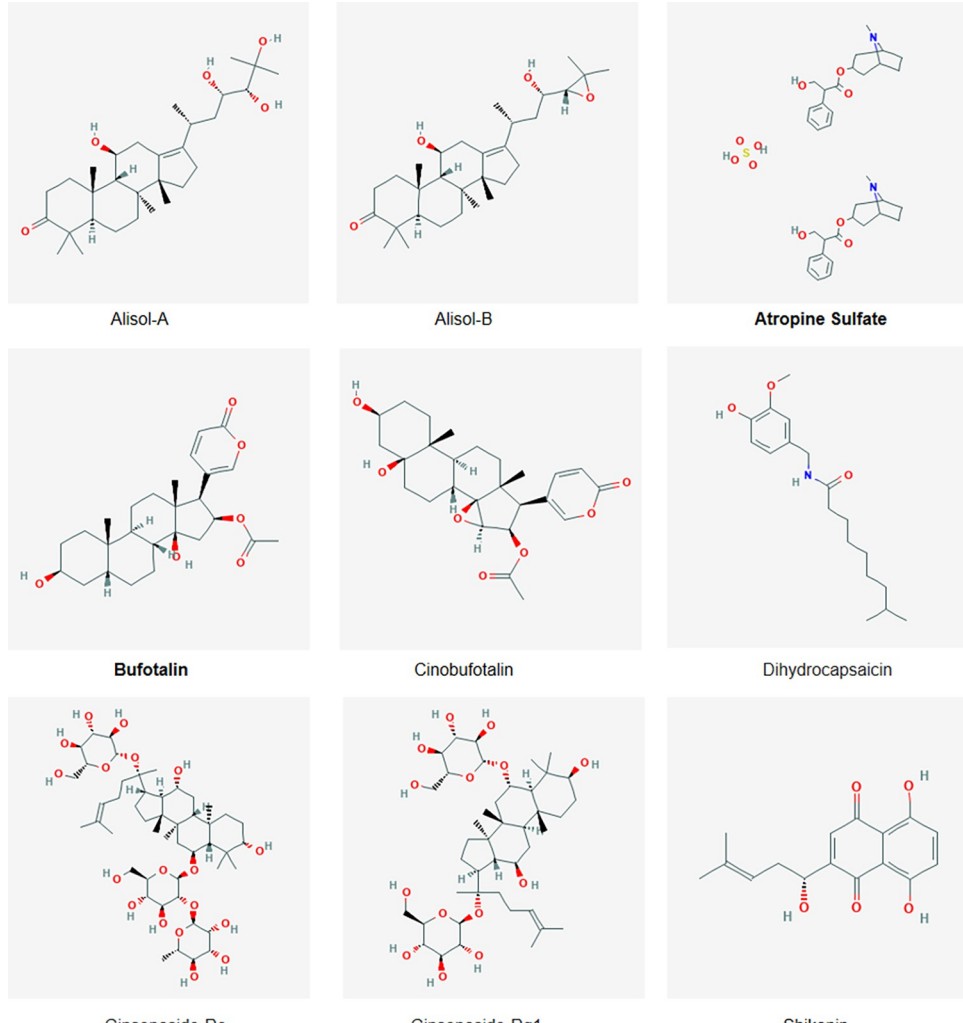

**Fig 2. Chemical structures of the hit compounds.** The nine compounds identified in the primary *in vitro* screen using 1 μM as the highest concentration. The most effective compounds are indicated by boldface letters. These compound structures were obtained from PubChem (https://pubchem.ncbi.nlm.nih.gov).

treatment with alisol-A or alisol-B led to no notable reduction of oocysts (S2 Fig). We, therefore, focus on the other two compounds. Atropine sulfate at 200 mg/kg showed modest efficacy (67.8% inhibition) against *C. parvum*, and bufotalin demonstrated significant inhibitory activity (78.1% inhibition) at 0.1 mg/kg and led to a rapid reduction in the number of oocysts in feces compared to the control group (Fig 4 and Table 3).

## Histopathological observations with immunohistochemistry

On examining a part of the small intestine, *C. parvum* infection and inflammation was most prominent in the ileum. In the positive control group (non-treated group), thickening of the mucosal layer, interstitial fibrosis, and inflammatory cell infiltration were observed (Fig 5D and 5E; the severity of *C. parvum* infection; Score: 3+) compared to the negative control group (Fig 5A–5C; Score: -). *C. parvum* oocysts were observed as tiny round basophilic organisms in the Hematoxylin & Eosin (HE), attached to the surface of the small intestinal villi of the intestinal crypt (Fig 5E, 5H, 5K and 5N). In immunohistochemical experiments, *C. parvum* oocyst

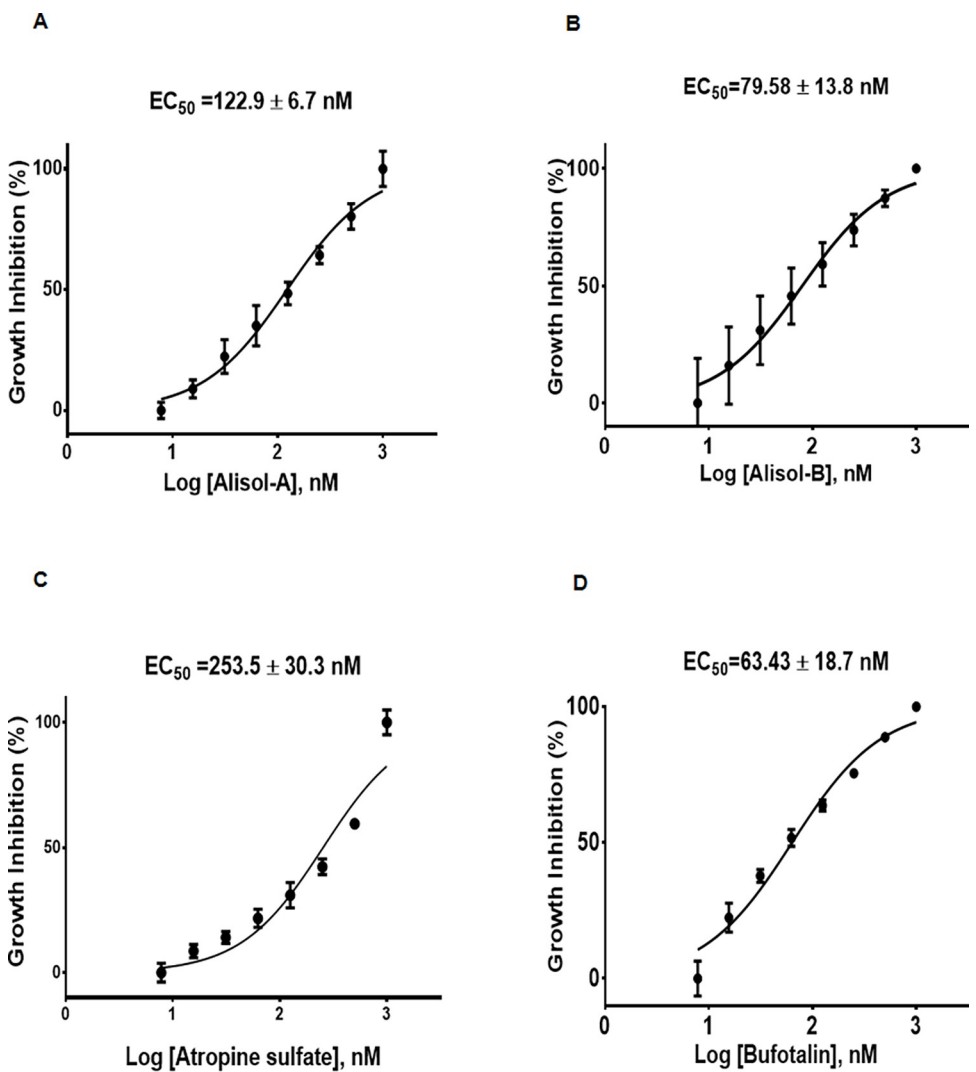

**Fig 3. Dose-response curves and growth inhibitory effects of four hit compounds on *C. parvum*.** The growth of *C. parvum* HNJ-1 strain after exposure to various concentrations of four hit compounds for 45 h, as determined by means of fluorescence microscopy. The EC$_{50}$s were determined from dose response curves using non-linear regression (curve fit analyses). The values from triplicate experiments are shown. (A) Dose-response curves and the half maximum inhibition concentration (EC$_{50}$) value of alisol-A for *C. parvum* (122.9± 6.7 nM), (B) Dose-response curves and the (EC$_{50}$) value of alisol-B (79.58±13.8 nM), (C) Dose-response curves and the (EC$_{50}$) value of atropine sulfate (253.5 ±30.3 nM), (D) Dose-response curves and the (EC$_{50}$) value of bufotalin (63.43±18.7 nM).

showed a positive reaction and were clearly distinct from those observed in the HE (Fig 5F, 5I, 5L and 5O). For the nitazoxanide-treated groups, a number of attached *C. parvum* oocysts were also observed in the ileum (Fig 5G–5I; Score: 2+). In the atropine-sulfate-treated groups, *C. parvum* infection led to a smaller number of oocysts being attached to the intestinal villi than in the positive control group (Fig 5J–5L; Score: 2+). Bufotalin-treated mice showed a significant reduction in the number of oocysts compared to the positive control group (Fig 5M–5O; Score: 1+). These results indicate that bufotalin is the most effective of the compounds tested in terms of anti-*Cryptosporidium* activity *in vivo*. Some histological findings (karyomegaly, hepatocellular hypertrophy, or interstitial inflammation) were observed in all examined livers (S3 Fig), it suspected sub-effect of compounds or infection, but there was no difference in distribution and severity between experimental groups.

**Table 2. Biological activity of hit compounds evaluated against the *in vitro* growth of *C. parvum*.**

| Compound | EC$_{50}$ value (nM)[a] | CC$_{50}$ (μM)[b] | 95%CI[c] | Selectivity index[d] |
|---|---|---|---|---|
| Alisol-A | 122.9±6.7 | 59.6 | 104.9–144.0 | 484.4 |
| Alisol-B | 79.58±13.8 | 36.7 | 61.0–103.8 | 460.6 |
| Atropine sulfate | 253.5±30.3 | 17.7 | 205.1–312.4 | 69.9 |
| Bufotalin | 63.43±18.7 | 10.9 | 54.5–73.9 | 172.3 |

Growth inhibitory effects ([a]EC$_{50}$ values) of lead compounds were evaluated *in vitro* by using a fluorescence assay in three separate experiments. Each compound concentration was tested in triplicate in each experiment, and the final EC$_{50}$ values are the mean values obtained from the three separate experiments.

[b] The CC$_{50}$ values on HCT-8 cells.

[c] 95%CI, 95% confidence intervals for EC$_{50}$ values.

[d] Selectivity indices were calculated based on the ratio of CC$_{50}$ /EC$_{50}$ for each compound.

## Scanning electron microscopy (SEM)

Next, we looked to see if there were any differences in the effects of the TCM compounds on host cells and tissues following *C. parvum* invasion. To achieve this, we examined the intestines of mice infected with *C. parvum* under SEM. SEM observation revealed ileal tissues on the surface of the intestine mucosa after infection with *C. parvum* and treatment with the TCM compounds. No abnormalities were detected in the intestinal epithelial tissues except that we found parasite oocysts attached to the ileal mucosa in mice treated with the comparative drug nitazoxanide (Fig 6G–6I). However, atropine sulfate- (Fig 6J–6L) and bufotalin (Fig 6M–6O)-

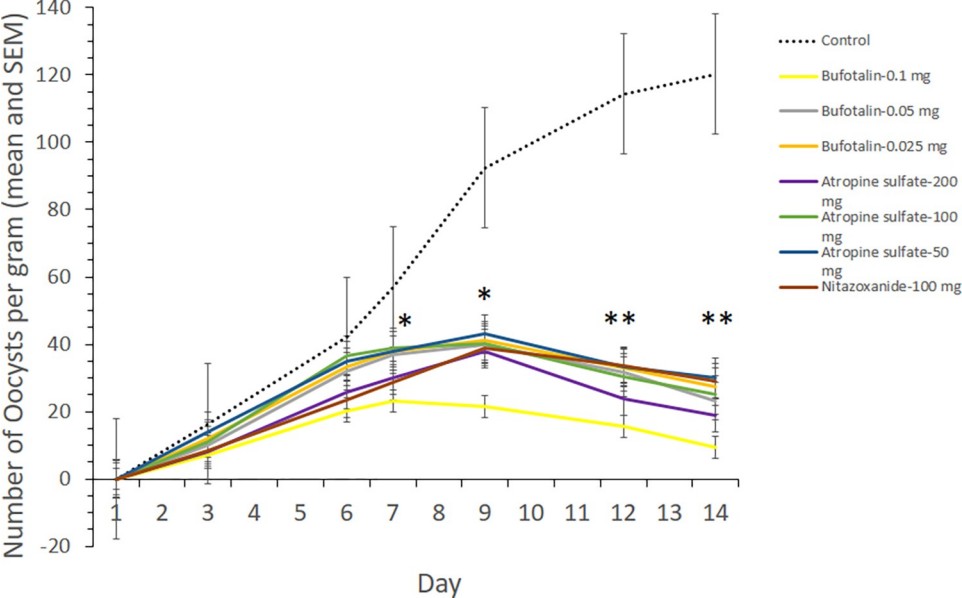

**Fig 4. *In vivo* infected mice treated with compounds showed growth inhibition of *C. parvum*.** Two compounds (atropine sulfate and bufotalin) reduce the number of oocysts shedding in mice compared to control drugs. *C. parvum* oocysts were inoculated orally into SCID mice. Infected mice left untreated served as a positive control. The mice were treated with 200 μl of atropine sulfate or bufotalin 3 days after infection (i.e., from Day 3) until Day 13 at three different concentrations for each compound (atropine sulfate: 200, 100, and 50 mg/kg; bufotalin: 0.1, 0.05, 0.025 mg/kg). Nitazoxanide was administered orally at 100 mg/kg as a comparative drug. The number of oocysts in the feces was determined by using the sugar flotation method, and the total number of oocysts per gram (OPG) was calculated. Data shown are the mean and SEM (n = 3 mice in each group). Asterisks indicate levels of statistical significance as evaluated by the difference in parasitemia between the control and drug-treated groups by use of a one-way ANOVA with the post-hoc Tukey HSD test, *: $p < 0.05$, **: $p < 0.01$.

**Table 3. Efficacies of TCM compounds with nitazoxanide for reducing oocyst shedding from the distal colon of *C. parvum*-infected neonatal SCID mice.**

| Days of treatment | Treatment compounds with dose (mg/kg BW) | No. of mice | No. of oocysts | | | % reduction in oocyst shedding[c] | p-value[d] |
|---|---|---|---|---|---|---|---|
| | | | Mean[b] | SD | 95% CI | | |
| 11 | Atropine sulfate 200 | 3 | 71.1 | 31.1 | 96.1–46.2 | 67.8 | 0.017* |
| 11 | Atropine sulfate 100 | 3 | 91.1 | 33.3 | 117.8–64.4 | 58.7 | 0.034* |
| 11 | Atropine sulfate 50 | 3 | 96.6 | 29.8 | 120.5–72.7 | 56.2 | 0.039* |
| 11 | Bufotalin 0.1 | 3 | 48.5 | 20.1 | 64.6–32.3 | 78.1 | 0.007** |
| 11 | Bufotalin 0.05 | 3 | 86.6 | 32.6 | 112.7–60.5 | 60.8 | 0.029* |
| 11 | Bufotalin 0.025 | 3 | 92.3 | 30.9 | 117.1–67.5 | 58.2 | 0.034* |
| 11 | Nitazoxanide 100 | 3 | 81.1 | 31.6 | 106.4–55.8 | 63.3 | 0.024* |
| | Control[a] | 3 | 221.1 | 125.4 | 321.5–120.7 | 00.0 | |

[a] Untreated mice infected with HNJ-1 strain *C. parvum* oocysts.

[b] The mean number of oocysts that shed on days of 3, 6, 7, 9, 12, 14.

[c] Microsoft excel was used to calculate the percent inhibition of infection as follows: 1-(mean number of parasites in drug treated group/mean number of parasites in control group) X 100.

[d] Pairwise comparisons using a one-way ANOVA with the post-hoc Tukey HSD test were used to determine whether reductions in numbers of oocysts for treated vs. untreated control mice were statistically significant. Results were considered to be significant for

*$P < 0.05$ or **$P < 0.01$.

treated mice showed no visible parasites in their intestine relative to the control groups (Fig 6A–6F), thereby confirming our findings in this study (Fig 6).

## Discussion

TCMs have a great variety of chemical structures, as well as extremely powerful pharmacological activity and relatively mild toxicities [25]. TCMs include a huge number of active chemicals, some of which have been used to generate novel medications to treat important diseases such as vascular disease and cancer [26]. Infection and transmission of *Cryptosporidium* parasites are difficult to manage; new medications to control infection and decrease the growth of apicomplexan parasites are needed. In this study, we found that atropine sulfate and bufotalin displayed the most anti-*Cryptosporidium* activity of 87 TCM compounds investigated. Bufotalin is widely used in anticancer therapy [27]. Previously, these TCM compounds were tested against *Plasmodium falciparum* and four (berberine chloride, coptisine chloride, palmatine chloride, and dehydrocorydaline nitrate) were found to have anti-malarial activity [17]. Moreover, TCM compounds have been tested against *Toxoplasma gondii* [18], with baicalein and luteolin being shown to have potential as anti-toxoplasmosis drugs. Notably, these TCM compounds that were effective in earlier studies were not effective against cryptosporidiosis. It is surprising that there is no overlap in the hits vs. *Plasmodium* and *Toxoplasma*, given the related of these species with *Cryptosporidium*. It might be due to the different enzyme related to metabolic cyclic pathways or functions related to either host or parasite factors. Our study is thus the first to reveal the inhibitory effects of atropine sulfate and bufotalin against *Cryptosporidium* parasite *in vitro* and *in vivo*.

Alisol-A, alisol-B, atropine sulfate, and bufotalin have low $EC_{50}$ values, implying that they could be useful in developing a new anti-cryptosporidiosis compounds. All four compounds were also shown to have minimum cytotoxicity in human intestinal cells (HCT-8). Earlier report mentioned that bufotalin has modest cytotoxicity in mammals and was active against trypanosomiasis [28]. Also, bufotalin effectively suppressed the growth of xenografted R-HepG2 cells in an *in vivo* investigation, with minimal body weight loss or spleen toxicity

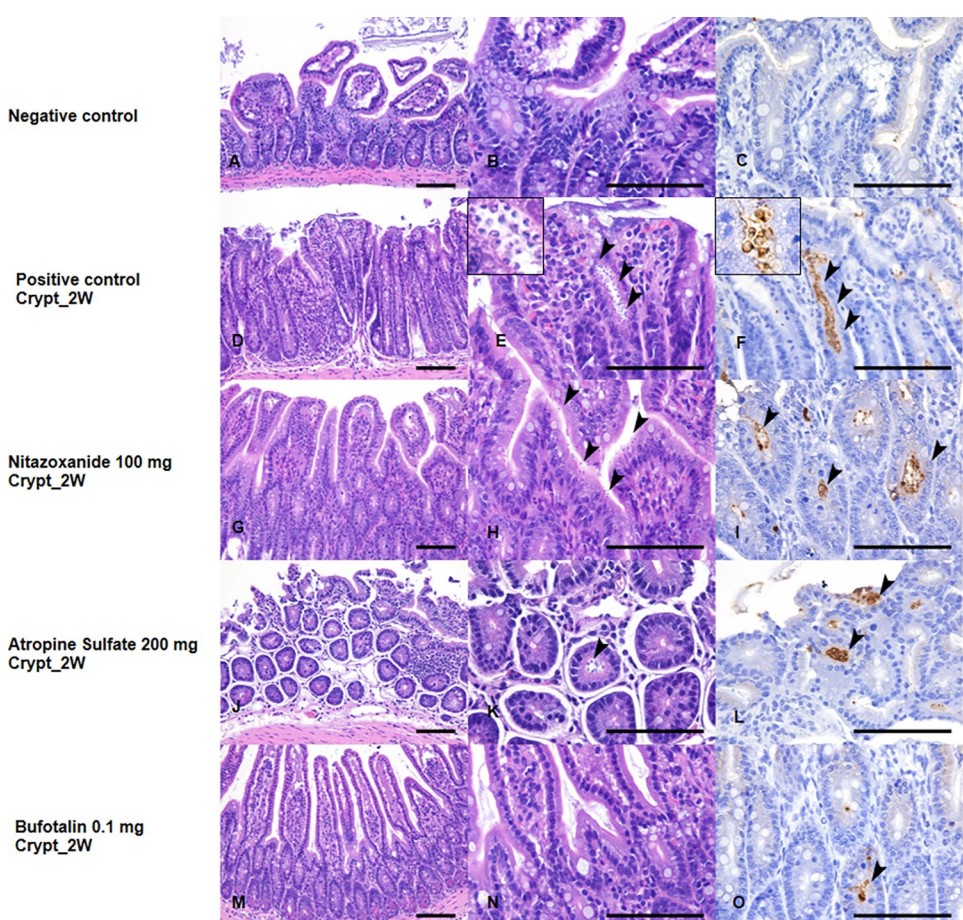

**Fig 5. Histological observation with immunohistochemistry detection of *C. parvum* infection in ileum tissues.** Histological sections of small intestine (ileum) of different animal groups (treated with nitazoxanide, atropine-sulfate, bufotalin, and control groups). Except for the negative control group, all mice were orally inoculated with *Cryptosporidium parvum* oocysts ($1 \times 10^5$), then treated with each compound. The left column is the lower magnification for hematoxylin and eosin (HE), the middle column is the higher magnification for HE, and the right column is immunohistochemistry for *C. parvum*. The severity of *C. parvum* infection was scored from– to 3+. (A-C) Ileal sections of uninfected mouse (negative control) (Score: -). (D-F) Ileal sections of non-treated mouse (positive control) showing inflammatory cell infiltration of the mucosal layer. Numerous *C. parvum* oocysts are attached to the intestinal villi (Score: 3+) (arrowheads: *C. parvum* oocysts). Ileal sections of nitazoxanide-treated groups (G-I), Nitazoxanide-treated group (Score: 2+) showing a number of oocysts. (J-L) Atropine sulfate-treated group showing less reduction of oocysts (Score: 2+). (M-O) Bufotalin-treated group showing markedly reduced number of oocysts (Score: 1+). Bar = 100 μm.

[29]. With an IC$_{50}$ of 3.7 μM against *Cryptosporidium parvum*, nitazoxanide is the only limited-activity medication currently approved by the FDA for the treatment of cryptosporidiosis in humans [30]. Atropine sulfate and bufotalin had a strong inhibitory impact *in vitro* but appeared to have a considerable inhibitory effect *in vivo* in SCID mice, and bufotalin had the greatest inhibitory effect. In this study, bufotalin showed a significant reduction in oocyst excretion at a concentration of 0.1 mg/kg, whereas atropine sulfate showed a moderate reduction in oocyst excretion at a concentration of 200 mg/kg. Moreover, bufotalin had potent inhibitory effects at lower doses than nitazoxanide. It is unclear why alisol-A and alisol-B inhibit *C. parvum* growth *in vitro* but have no effect *in vivo*. One of the factors might be the lack of adequate intestinal exposure to these drugs. Because it is likely that prolonged intestinal exposure of these TCMs is crucial for the efficacy *in vivo*. This is important because it may be

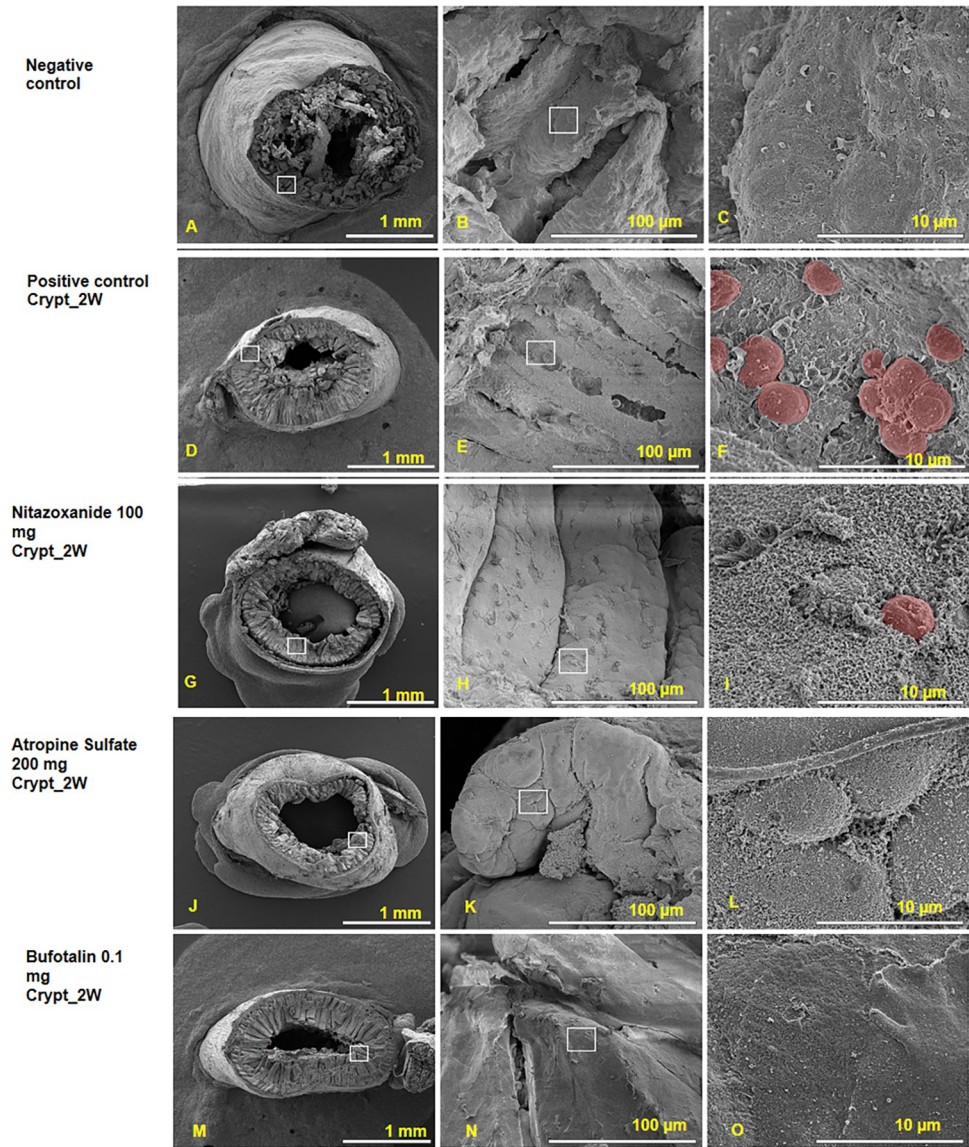

**Fig 6. Scanning electron microscopic (SEM) findings the compound's effects of ileum tissues.** The SEM images of intestinal ileum tissue of different groups of mice infected with *C. parvum* and treated with hit compounds, as well as the negative (A-C) and positive (D-F) control groups, and comparative drug (nitazaxonide)-treated group (G-I). Atropine sulfate- (J-L) and bufotalin- (M-O) treated mice showed no abnormalities or visible parasites in the intestine. The SEM shows the intracellular structures as well as the surface of *C. parvum* attached to the host cells. These compounds had no adverse effects on the ileum tissue.

that modifications of alisol-A and alisol-B would produce compounds with better intestinal exposure and *in vivo* efficacy. The nature of the chemical compounds, whether the drug compounds are permeable or less permeable to the small intestine would be the other factors [31]. However, variables such as host cell metabolism or the parasite's capacity to target only specific molecules could restrict the compound's effectiveness.

Although the molecular mechanisms of the anti-*Cryptosporidium* activity of TCMs remain unclear, they may target enzyme-related molecules. Earlier studies reported, according to an ADME (absorption, distribution, metabolism, and excretion) and toxicity study, atropine crosses the blood-brain barrier, which is crucial for efficient treatment of viral illness [32].

Moreover, submicronic atropine sulfate respiratory fluid has the potential to be employed as a prophylactic strategy against organophosphorous poisoning, with various advantages over intramuscular injection, including early blood absorption and atropinization [33]. Calcium-dependent protein kinases play important roles in calcium signaling in *T. gondii* and *C. parvum*, making them interesting targets for antiparasitic drug research [34, 35]. In addition, the regulation of calcium ions is required for the activity of calcium-dependent protein kinases (CDPKs), which are found in *C. parvum* [36]. Furthermore, alisol-B is known to inhibit the endoplasmic reticulum $Ca^{2+}$ ATPase [37], and most believe that nitazoxanide anticryptosporidial effect is off-target, as *Cryptosporidium* species has an alternative PFOR that nitazoxanide and more potent PFOR inhibitors have no effect *in vitro* [38]. It may be that atropine sulfate and bufotalin suppress *Cryptosporidium* via a different target(s) than that of nitazoxanide.

Bufotalin, one of the naturally occurring bufodienolides, has pharmacological and toxicological features that include anticancer activity and cardiotoxicity [39]. Bufotalin was found to be less harmful to Het-1A human esophageal squamous cells, implying that it has a high selectivity for cancer cells [40]. Atropine sulfate was found to have a considerable inhibitory impact and bufotalin had a potent inhibitory impact, which could be attributed to blocking a parasite metabolic route or the targeting of several enzymes in host cell-dependent pathways. Our future studies will explore the mechanisms of these compounds against enzyme-related molecules because it is unknown if these compounds are working by virtue of inhibiting *Cryptosporidium* factors or host factors.

In conclusion, our findings suggest that the compounds atropine sulfate and bufotalin could be useful in the development of new anti-*Cryptosporidium* medications. More research is needed, however, to understand the exact mechanism behind the anti-*Cryptosporidium* activity of these compounds.

## Supporting information

**S1 Fig. The Effect of TCMs treated compounds on the body weight during *C. parvum* infection in mice.** The body weights of mice infected with *C. parvum* are shown with the following drugs. Control: Infected-untreated mice, Alisol-A: Infected mice treated with different concentration of alisol-A at 50, 25, and 12.5 mg/kg for 11 consecutive days, Alisol-B: Infected mice treated with different concentration of alisol-B at 50, 25, and 12.5 mg/kg for 11 consecutive days, Atr-S: Infected mice treated with different concentration of atropine-sulfate at 100, 50, 25mg/kg for 11 consecutive days, Bufotalin: Infected mice treated with different concentration of bufotalin at 1, 0.5, and 0.1 mg/kg for 11 consecutive days. Treated mice values were not significant compared with those of uninfected mice.
(TIF)

**S2 Fig. Histological and SEM observation of *C. parvum* infected ileum tissues treated with compounds.** (A) Histological sections of the small intestine (ileum) of different animal groups (treated with alisol-A and alisol-B). All mice were orally inoculated with *Cryptosporidium parvum* oocyst ($1\times10^5$), then treated with each compound. The severity of *C. parvum* infection was scored from–to 3+. Ileal sections of treated groups [Alisol-A (Score: 3+) and Alisol-B (Score: 2+)] showing no remarkable reduction in the number of oocysts (arrowheads: *C. parvum* oocysts). HE. Bar = 100 μm. (B) Scanning electron microscopic (SEM) images of intestinal ileum tissue of two groups of mice infected with *C. parvum*. The SEM shows the intracellular structures as well as the surface of *C. parvum* attached to the host cells.
(TIF)

**S3 Fig. Histological observation of *C. parvum*-infected liver tissues treated with compounds.** The higher magnification for the histological sections of the liver of different animal groups. Except for the negative control group, all mice were orally inoculated with *Cryptosporidium parvum* oocysts ($1\times10^5$), then treated with each compound. (A) Uninfected mouse (negative control) showing karyomegaly and hepatocellular hypertrophy. (B) Non-treated mouse (positive control) showing hepatocellular hypertrophy and inflammation of the hepatocytes. (C) Nitazoxanide-treated groups showing karyomegaly and hepatocellular hypertrophy. (D) Atropine sulfate-treated group showing inflammation. (E) Bufotalin-treated group showing also inflammation and hepatocellular hypertrophy. Hematoxylin and eosin (HE) stain, Bar = 50 μm.
(TIF)

**S1 Table. Comparative efficacies of four hit compounds treated for reducing oocysts shedding of *C. parvum*-infected neonatal SCID mice.**
(DOCX)

**S1 Data. Excel spreadsheet containing the underlying numerical data and statistical analysis for Fig 1.**
(XLSX)

## Acknowledgments

We thank the Institute of Natural Medicine (The University of Toyama, Toyoma, Japan) for providing the chemical compounds. We thank Dr. M. Matsubayashi (Osaka Prefecture University, Japan) for providing the *C. parvum* HNJ-1 strain used in this study.

## Author Contributions

**Conceptualization:** Mohammad Hazzaz Bin Kabir, Kentaro Kato.

**Data curation:** Mohammad Hazzaz Bin Kabir, Frances Cagayat Recuenco, Nur Khatijah Mohd Zin.

**Formal analysis:** Mohammad Hazzaz Bin Kabir, Kenichi Watanabe, Hiroki Bochimoto.

**Funding acquisition:** Mohammad Hazzaz Bin Kabir, Kentaro Kato.

**Investigation:** Mohammad Hazzaz Bin Kabir, Nina Watanabe, Yasuhiro Fukuda, Hironori Bando, Kenichi Watanabe, Hiroki Bochimoto.

**Methodology:** Mohammad Hazzaz Bin Kabir.

**Project administration:** Xuenan Xuan, Kentaro Kato.

**Resources:** Kentaro Kato.

**Supervision:** Xuenan Xuan, Kentaro Kato.

**Validation:** Mohammad Hazzaz Bin Kabir, Kentaro Kato.

**Visualization:** Mohammad Hazzaz Bin Kabir, Kentaro Kato.

**Writing – original draft:** Mohammad Hazzaz Bin Kabir.

**Writing – review & editing:** Mohammad Hazzaz Bin Kabir, Hironori Bando, Kenichi Watanabe, Hiroki Bochimoto, Kentaro Kato.

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
