## [Decision Letter · Decision Letter 0]

10 Aug 2022

Dear Dr. Kato,

Thank you very much for submitting your manuscript "Identification of potent anti-Cryptosporidium drug candidates by screening traditional Chinese medicines" for consideration at PLOS Neglected Tropical Diseases. As with all papers reviewed by the journal, your manuscript was reviewed by members of the editorial board and by several independent reviewers. In light of the reviews (below this email), we would like to invite the resubmission of a significantly-revised version that takes into account the reviewers' comments. 

We cannot make any decision about publication until we have seen the revised manuscript and your response to the reviewers' comments. Your revised manuscript is also likely to be sent to reviewers for further evaluation.

Sincerely,

Richard Stewart Bradbury, PhD

Academic Editor

Alain Debrabant, PhD

Section Editor

Reviewer's Responses to Questions

**Key Review Criteria Required for Acceptance?**

**Methods**

-Are the objectives of the study clearly articulated with a clear testable hypothesis stated?

-Is the study design appropriate to address the stated objectives?

-Is the population clearly described and appropriate for the hypothesis being tested?

-Is the sample size sufficient to ensure adequate power to address the hypothesis being tested?

-Were correct statistical analysis used to support conclusions?

-Are there concerns about ethical or regulatory requirements being met?

Reviewer #1: -Are the objectives of the study clearly articulated with a clear testable hypothesis stated? YES

-Is the study design appropriate to address the stated objectives? Some issues - detailed in review

-Is the population clearly described and appropriate for the hypothesis being tested? N/A

-Is the sample size sufficient to ensure adequate power to address the hypothesis being tested? YES

-Were correct statistical analysis used to support conclusions? YES

-Are there concerns about ethical or regulatory requirements being met? NO

Reviewer #2: 1) objectives are clear

2) study design is appropriate

3) sample size may be too small (3 mice per group in final efficacy study)

4) one-way ANOVA with correction was used and is appropriate--they need to show the statistics in table 3 in addition to stars in figure 4

5) no ethical concerns

Reviewer #3: For the analysis of the data in figure 4, a more appropriate statistical test would be repeated measures ANOVA (also available in GraphPad). This test compares the overall curves of each group rather than the data at different days. It is highly unlikely that there will be any difference in outcomes using this test, but it is just a better choice for the type of experiment performed.

Reviewer #4: In the methods, this reviewer has the following concerns for the authors to consider:

1) The hit rate is quite high at 1 μM (9 out of 87 compounds), which is uncommon for a random collection of natural products. It is necessary to ensure that the phenotypic HTS assay used in the study is reliable. 

For the in vitro screen and efficacy assays, fluorescence-imaging based quantification assay using Sporo-Glo (that also labels intracellularly developing C. parvum) is not one of the well-established assays. There is a need to show the quality of the assay, including: 1) standard curves (signal intensity vs. # of parasite inoculums) to show the linearity and linear dynamic range of the assay, and 2) dose-dependent curve of a standard control compound for comparison (e.g., NTZ or PRM). 

There is also a need to describe how the fluorescence data were analyzed. Some representative images of the control and treated specimens will also be helpful in assessing the assay quality.

For the compound library, there is a need to give a brief description on the background of the 87-compound library, and citations if available.

2) In animal experiments using SCID mice:

A) Only female mice were used, and the oocyst shedding in SCID mice (max. at ~120 oocysts/gram of feces) was much lower than previous reported (typically in mg scale). The authors may want to add some discussion on the fact, and mention that there might be a need to further validate the in vivo efficacy in other models with both male and female animals. 

B) Please clarify some important technical details, including: 1) how fecal pellets were collected from individual mice (e.g., all pellets dropped in past 24 h or fresh pellets dropped in the past 1 h); and 2) how oocysts were counted/quantified after sugar flotation method, plus critical details of the “sugar flotation method”. 

3) For the parasite strain (C. parvum strain HNJ-1): Please provide a GP60-based genotype/subtyping info (as an important parameter for comparison with other data in the literature).

**Results**

-Does the analysis presented match the analysis plan?

-Are the results clearly and completely presented?

-Are the figures (Tables, Images) of sufficient quality for clarity?

Reviewer #1: -Does the analysis presented match the analysis plan? Mostly YES

-Are the results clearly and completely presented? NO

-Are the figures (Tables, Images) of sufficient quality for clarity? YES

Reviewer #2: 1) analysis presented is appropriate

2) results are clearly presented but the authors have not yet made all data available, which they state will be done upon acceptance. More information re statistics is needed in Table 3. The nature of the error bars in Figure 4 needs to be indicated in the figure legend.

3) the figures and tables are clear--as above, clarifications/additions are needed for table 3 and figure 4

Reviewer #3: (No Response)

Reviewer #4: The results were reasonably well presented. Here are a few comments for clarification:

1) For clarity, it is suggested to change IC50 (recommended for use of 50% inhibition on a drug target in biochemical assays) to EC50 (recommended for use of 50% inhibition on parasite growth in vitro).

2) Histological images: Some local high-resolution images showing the parasites can be helpful.

3) In Table 2: please define the parameter for the 95% CI (95% CI for EC50 or CC50 values?).

4) Primary screen data (anti-cryptosporidial activities in vitro of the 87 compounds at 1 μM) can be provided in spreadsheet as part of the supplementary materials.

**Conclusions**

-Are the conclusions supported by the data presented?

-Are the limitations of analysis clearly described?

-Do the authors discuss how these data can be helpful to advance our understanding of the topic under study?

-Is public health relevance addressed?

Reviewer #1: Are the conclusions supported by the data presented? Not completely

-Are the limitations of analysis clearly described? No

-Do the authors discuss how these data can be helpful to advance our understanding of the topic under study? Somewhat

-Is public health relevance addressed? Not completely

Reviewer #2: 1) The general conclusions are supported by the data. That said, it is an overstatement to indicate they have identified drug candidates in the title, abstract, lay summary and throughout the paper. That is not the case based on how candidate is generally used in drug development. They have identified new lead compounds.

2) limitations are described--editorial suggestions for additions indicated below

Reviewer #3: (No Response)

Reviewer #4: The conclusions are generally supported by the data presented.

**Editorial and Data Presentation Modifications?**

Reviewer #1: It is clear the authors first language is not English, it is not the worst written manuscript I have read but would benefit from someone reading and correcting the grammar prior to submission.

Reviewer #2: Editorial suggestions:

1) Candidates is an overstatement, as implies optimization and derisking have been completed. More appropriate drug leads--same in abstract and throughout the manuscript

2) Line 75 about clofazimine is out of date. This repurposing screening hit was tested in a clinical trial in HIV patients with cryptosporidiosis and it was ineffective (see Clinical Infect Dis. 2021. 73: 183-91 and Clinical Infect Dis. 2021. 73: 192-4.)

3) Line 82 indicating a lack of studies showing compounds in cattle that reduce clinical symptoms is incorrect. Many studies have now been published for compounds in development showing efficacy against diarrhea in dairy calves (see PMID 31249291, PMID 29309415, PMID 28562588, and PMID 27923949).

4) Line 134--use of nitazoxanide as comparative drug. Specify what concentration of nitazoxanide was used for this control. Was it 1 micromolar like the screening compounds? Nitazoxanide lacks potency and is inactive at 1 micromolar in our hands and also published literature.

5) Figure 4: please specify what the error bars are in the figure legend. Are these standard deviation or standard error?

6) Table 3: a column should be added or asterisks, etc and a definition to indicate the results of statistical analysis. Also, for calculation of percent inhibition, are these comparisons vs control for a single time point or for area under the curves shown in figure 4?

7) Discussion lines 306-309: It is surprising and a little concern that there is no overlap in the hits vs Plasmodium and Toxoplasma, given the related of these species with Cryptosporidium. While differences in PK/PD requirements for in vivo efficacy can easily account for in vivo differences, most studies show an enrichment of active compounds in vitro (e.g. screens that have been done using the MMV Malaria Box; PMID 24566188 and PMID 29339392). More discussion of this would be of interest.

8) Line 327: lack of in vivo efficacy for Alisol-A and -B. This discussion needs to be expanded some. There is considerable data at this point suggesting that in vivo efficacy vs. Cryptosporidium is dependent on prolonged intestinal exposure. Lack of adequate intestinal exposure is the likely explanation (see PMID 28541457). This is important because it may be that modifications of Alisol-A and -B would produce compounds with better intestinal exposure and in vivo efficacy.

9) Line 341 re mechansim of nitazoxanide: While nitazoxanide is a PFOR inhibitor, there is considerable evidence that that is not its mechanism of action against Cryptosporidium species. Cryptosporidium species have an alternative PFOR that nitazoxanide and more potent analogs don't inhibit (see PMID 30297368).

10) Line 348 re future studies of mechanism: you should note also that it is unknown if these compounds are working by virtue of inhibiting Cryptosporidium factors or host factors. There are many publications demonstrating dependence on host factors (e.g. protein kinase C (PMID 35099276) and aquaporin (PMID 15851691)).

Reviewer #3: Lines 24-25 and 44 are oddly worded--the authors likely intended to state that cryptosporidiosis is the 2nd most common diarrheal disease, with rotavirus being the most common. But as worded, it sounds like cryptosporidiosis only occurs after someone has rotavirus infection. 

The placement of a sentence about the anti-cancer effects of bufotalin in lines 77-78, between 2 sentences discussing compounds with anti-cryptosporidial activity is odd and at first gives the impression that bufotalin has previously been tested against cryptosporidia. It is probably best to move this sentence to the discussion.

Similarly, the sentence about bufotalin selectivity in lines 344-345 is oddly placed and breaks the flow of the paragraph. Perhaps the authors should create a separate paragraph discussing the effects of bufotalin in other studies.

Reviewer #4: 1) In reviewing anti-cryptosporidial drug discovery, there are several other good review articles and original research articles reporting phenotypic-based HTS. A more relevant one is the HTS of natural products (PubMed PMID: 31551955; PubMed Central PMCID: PMC6736568).

2) Line 23, Abstract, first sentence: “Cryptosporidium spp. are intestinal opportunistic protozoan parasites…”: change intestinal to gastrointestinal (as there are also gastric Cryptosporidium species such as C. muris).

3) Other language and grammatical issues:

Line 46 and many other places: Please make “Cryptosporidium” (anti-Cryptosporidium) and “C. parvum” in italic.

Line 67 and other places if any: change the first letter in “Nitazoxanide” to lowercase (nitazoxanide) unless it starts a sentence. The same applies to other compound names and terms such as “Paromomycin”, “Malaria”, “Toxoplasmosis” and “Cancer” (lines 89 -90). 

Some other irregular usages. For example, leave a space between a number and unit (e.g., line 164, 100μl could be 100 μL; line 172, the μ in μM does not need to be in italics …).

**Summary and General Comments**

Reviewer #1: The authors determine the effect of a series of compounds used in traditional Chinese medicine on the in vitro and in vivo growth of Cryptosporidium parvum. The results indicate the potential of bufotalin and possibly atropine sulfate for the chemotherapy of Cryptosporidosis, which currently lacks a robust chemotherapeutic strategy. Overall, the data is interesting and would be a useful guide to researchers in the field potentially leading to new chemotherapies for this neglected disease. In it’s present form there are many inconsistencies that should be addressed; some of these are grammatical and are easy fixes, but there are also some technical problems in the methodology that should be addressed prior to publication.

Specific comments:

1. Summary – line 46 change ‘top goal’ to priority

2. Summary – line 54 replace ‘also showed in vivo inhibitory effects’ with reduced oocyst shedding by 68% and 78%, respectively.

3. Summary – line 56-57 Could be good candidates … sounds as if the authors are uncertain of the potential better to state: The chemotherapeutic potential of these compounds are discussed.

4. Introduction – line 70 should add or children under 2 years after immunodeficient patients.

5. Methods – line 131 delete ‘To begin’

6. Methods – line 139 the composition and concentration of phosphate buffer should be stated.

7. Methods - line 140 similarly the composition and concentration of phosphate buffered saline should be stated, and the filter conditions described – what pore-size filter, syringe or pump driven, etc.

8. Methods – line 141 the HCT-8 cells were ‘treated’ with C. parvum? Do they mean ‘infected’

9. Methods – line 142 are the compounds in the 1 uM library dissolved in DMSO? The statement ‘uninvaded parasites’ is misleading and should be changed – the host cells are invaded? 

10. Methods – line 144 the description sounds like the compounds added twice? Is this correct.

11. Methods – line 145-147 The fixation step needs more details, is the medium removed prior to addition of methanol? Also the statement ‘the cells were blocked…’ is vague, what cells host, parasite? And Sporo-Glo primarily stains sporozoites that are extracellular, motile stages, which are presumably removed by the washing? Is this method staining intracellular meront stages if so an explanation is needed.

12. Methods line – 180 What are the C. parvum oocysts suspended in?

13. Methods line – 185 What unfavorable effects?

14. Methods line – 193-195 In mouse model infections oocyst shedding does not occur under 5 days? How were the oocysts detected, using a modified acid-fast stain, Crypt-a-Glo?

15. Methods line – 199 side effects should be changed to toxicity.

16. Methods line – 210-212 The opening statement to this section indicates evaluation of growth inhibition and toxicity bit it is not clear how will toxicity be evaluated using this method? 

17. Methods paragraph starting line – 214 A statement indicating what structural features are being looked at to evaluate toxicity should be added.

18. Results – why did the authors double the dose for atropine from 100 to 200 mg/kg when it is clear that 100 mg/kg did not cure? This is especially relevant since bufotalin was used at a fraction of this dose 0.1-0.025 mg/kg. With a mouse dose of 200 mg/kg it would scale to >10 g to treat a human? 

19. Results – The data in Table 2 should be at best to one decimal place, the confidence in two decimal places is not supported by the analysis used.

20. Results – since the in vitro method used to evaluate parasites fixes them, the autors do not know if what they detect are non-viable stages which can give misleading data. Do the authors know if parasites can recover after treatment, this could have been done by taking a sample from in vitro incubation with the compound for a fixed time and infecting mice. Alternatively using RT-PCR of parasite 18-S RNA which has a short half-life would be a better indicator of viable parasite load. The uncertainty of parasite viability post compound treatment might explain some of the differences between in vitro and in vivo data obtained.

Reviewer #2: This manuscript from Hazzaz Bin Kabir reports screening and initial in vitro and in vivo follow up of a collection of compounds based on traditional Chinese medicines against Cryptosporidium parvum. These compounds haven't previously been screened against Cryptosporidium and several new leads with activity in a mouse model of infection were identified. The methods used are well established and standard, and the work appears to have been executed well. Overall, the results are of interest, as additional lead compounds are needed to develop better drugs for cryptosporidiosis. The importance of the results is somewhat over-represented by use of the word "candidate" as far more work is needed before any of these compounds would be considered candidates, which implies a far more thorough evaluation regarding PK, dose-scaling, and toxicology. That said, the compounds do represent interesting new leads worthy of further evaluation. I have additional fairly minor editorial suggestions as noted above.

Reviewer #3: (No Response)

Reviewer #4: In this manuscript, Bin Kabir et al. reported the identification of anti-cryptosporidium activity of four compounds from traditional Chinese medicine herbs in vitro and in vivo (SCID mice), i.e., alisol-A, alisol-B, atropine sulfate and bufotalin. The four hits showed excellent, nanomolar anti-cryptosporidial activities (EC50 = ~80 to ~250 μM). Two compounds (atropine sulfate and bufotalin) were further evaluated in vivo, showing efficacy against the infection of C. parvum (reduction of oocyst shedding). 

These observations are very interesting and important by providing new structures for potentially further development of anti-cryptosporidial therapeutics.

PLOS authors have the option to publish the peer review history of their article (what does this mean?). If published, this will include your full peer review and any attached files.

Reviewer #1: Yes: Nigel Yarlett

Reviewer #2: No

Reviewer #3: No

Reviewer #4: Yes: Guan Zhu
---

## [Decision Letter · Decision Letter 1]

14 Nov 2022

Dear Dr. Kato,

We are pleased to inform you that your manuscript 'Identification of potent anti-Cryptosporidium new drug leads by screening traditional Chinese medicines' has been provisionally accepted for publication in PLOS Neglected Tropical Diseases.

Best regards,

Richard Stewart Bradbury, PhD

Academic Editor

Alain Debrabant, PhD

Section Editor

Reviewer's Responses to Questions

**Key Review Criteria Required for Acceptance?**

**Methods**

-Are the objectives of the study clearly articulated with a clear testable hypothesis stated?

-Is the study design appropriate to address the stated objectives?

-Is the population clearly described and appropriate for the hypothesis being tested?

-Is the sample size sufficient to ensure adequate power to address the hypothesis being tested?

-Were correct statistical analysis used to support conclusions?

-Are there concerns about ethical or regulatory requirements being met?

Reviewer #1: The objectives are clearly stated; the study design is appropriate; the in vitro and in vivo experiments are appropriate for the study; the sample size is sufficient for statistical analysis of the results. No ethical concerns.

Reviewer #2: Objectives: yes

design appropriate: yes

population: na

sample size: saw statistically significant effects, so sample size with adequate power. It appears only one mouse experiment was performed, which is a concern re reproducibility and rigor, e.g. if something as simple as mislabeling tubes occurred.

analysis appropriate: yes

no ethical concerns

Reviewer #4: Authors have addressed the concerns on the Methods in the revised manuscript.

**Results**

-Does the analysis presented match the analysis plan?

-Are the results clearly and completely presented?

-Are the figures (Tables, Images) of sufficient quality for clarity?

Reviewer #1: The analysis is consistent with the proposed study; The results are clearly presented; The Figs are of a sufficient quality.

Reviewer #2: yes, analysis matches plan

clearly presented

figures, tables, etc are clear

Reviewer #4: Authors have addressed the concerns.

**Conclusions**

-Are the conclusions supported by the data presented?

-Are the limitations of analysis clearly described?

-Do the authors discuss how these data can be helpful to advance our understanding of the topic under study?

-Is public health relevance addressed?

Reviewer #1: The conclusions are supported by the data presented; the limitations are not discussed but I do not consider this an issue for this study where the study design is consistent with current methods;The authors discuss the relevance of their findings with respect to the development of new therapies for this disease. Relevance to public health is discussed.

Reviewer #2: Yes, these issues all in-line in the modified manuscript

Reviewer #4: The conclusions are supported by the presented data.

**Editorial and Data Presentation Modifications?**

Reviewer #1: One minor comment, on Line 69: add abbrev (NTZ) after nitazoxanide.

Reviewer #2: see below

Reviewer #4: (No Response)

**Summary and General Comments**

Reviewer #1: The authors have addressed the comments satisfactorily. The manuscript will be a useful addition to the literature on drug development for the treatment of cryptosporidosis.

Reviewer #2: The authors adequately addressed all of my prior concerns in this revised manuscript. The conclusions and discussion are in-line with the data presented.

Reviewer #4: The manuscript has been revised to the satisfaction of this reviewer.

PLOS authors have the option to publish the peer review history of their article (what does this mean?). If published, this will include your full peer review and any attached files.

Reviewer #1: **Yes: **Nigel Yarlett

Reviewer #2: No

Reviewer #4: **Yes: **Guan Zhu

---

## [Editor Report · Acceptance letter]

25 Nov 2022

Dear Dr. Kato,

We are delighted to inform you that your manuscript, "Identification of potent anti-Cryptosporidium new drug leads by screening traditional Chinese medicines," has been formally accepted for publication in PLOS Neglected Tropical Diseases.

Best regards,

Shaden Kamhawi

co-Editor-in-Chief

Paul Brindley

co-Editor-in-Chief
